# RAD51 and RAD51B Play Diverse Roles in the Repair of DNA Double Strand Breaks in *Physcomitrium patens*

**DOI:** 10.3390/genes14020305

**Published:** 2023-01-24

**Authors:** Karel J. Angelis, Lenka Záveská Drábková, Radka Vágnerová, Marcela Holá

**Affiliations:** 1Institute of Experimental Botany, Czech Academy of Sciences, v.v.i., Na Karlovce 1, 16000 Prague 6, Czech Republic; 2Institute of Experimental Botany, Czech Academy of Sciences, v.v.i., Rozvojová 263, 16502 Prague 6, Czech Republic

**Keywords:** Physcomitrella, homologous recombination (HR), non-homologous end-joining (NHEJ), DNA double-strand break (DSB), repair kinetic, comet assay, rDNA, bleomycin, evolutionary divergence

## Abstract

RAD51 is involved in finding and invading homologous DNA sequences for accurate homologous recombination (HR). Its paralogs have evolved to regulate and promote RAD51 functions. The efficient gene targeting and high HR rates are unique in plants only in the moss *Physcomitrium patens* (*P. patens*). In addition to two functionally equivalent *RAD51* genes (*RAD1-1* and *RAD51-2*), other *RAD51* paralogues were also identified in *P. patens*. For elucidation of RAD51’s involvement during DSB repair, two knockout lines were constructed, one mutated in both *RAD51* genes (*Pprad51-1-2*) and the second with mutated *RAD51B* gene (*Pprad51B*). Both lines are equally hypersensitive to bleomycin, in contrast to their very different DSB repair efficiency. Whereas DSB repair in *Pprad51-1-2* is even faster than in WT, in *Pprad51B*, it is slow, particularly during the second phase of repair kinetic. We interpret these results as *PpRAD51-1* and -*2* being true functional homologs of ancestral *RAD51* involved in the homology search during HR. Absence of RAD51 redirects DSB repair to the fast NHEJ pathway and leads to a reduced 5S and 18S rDNA copy number. The exact role of the RAD51B paralog remains unclear, though it is important in damage recognition and orchestrating HR response.

## 1. Introduction

RAD51, the eukaryotic homolog of the bacterial recombinase RecA [1,2], plays a central role in homologous recombination (HR) in eukaryotes. Loss of *RAD51* function causes lethality in vertebrates, but not in other animals or in the flowering plant *Arabidopsis thaliana*, suggesting that *RAD51* is vital for highly developed organisms [3].

*RAD51* and its gene family are key regulators of DNA fidelity through the diverse roles in double-strand break (DSB) repair, replication stress, and meiosis. RAD51 is an ATPase that forms a nucleoprotein filament on single-stranded DNA. Assembly of RAD51 monomers onto ssDNA is a relatively slow process and is facilitated by several mediator proteins [4]. RAD51 has the function of finding and invading homologous DNA sequences to enable accurate and timely DNA repair. A family of proteins known as the RAD51 paralogs and consisting of five proteins (RAD51B, RAD51C, RAD51D, XRCC2, and XRCC3) play an essential role in the DNA repair reactions through HR [5]. The RAD51 paralogs act to transduce the DNA damage signal to effector kinases and to promote break repair. However, their precise cellular functions are still not fully elucidated [6]. Genes encoding these factors may have derived from ancestorial *RAD51* by gene duplication, and the corresponding proteins have subsequently acquired new activities [7]. Phylogenetic analyses indicated that the RecA/RAD51 family can be divided into subfamilies with a highly conserved or relatively divergent functions [7].

In the bryophyte *Physcomitrium patens* (*P. patens*), there are two RAD51 proteins, PpRAD51-1 and PpRAD51-2, which, like their counterparts from other organisms, show ATP-dependent filament formation and strand-exchange activity on DNA substrates. Respective genes evolved without introns by a recent duplication of the ancestral *RAD51* gene [8]. The *rad51-1-2* double mutant is viable, sterile, and shows a marked developmental phenotype and a strong hypersensitivity to the DSB-inducing agent bleomycin [9]. Bleomycin is a radiomimetic agent that induces a similar spectrum of DNA lesions as ionizing radiation [10]. *P. patens* is the only plant that naturally displays high HR rates and gene-targeting (GT) efficiencies [11]. Genetic studies showed that this feature is tightly associated with the *RAD51*-dependent HR repair pathway. Deletion of *RAD51* and *RAD51B* results in suppression of GT-driven homology in favor of an increase in untargeted, random integration of transgenes. Competition between HR and illegitimate (IR) recombination, which both underlie GT processes, suggests that the balance between HR-mediated and non-homologous end-joining (NHEJ) DSB repair may be shifted, at least in this organism [9,12]. Moreover, Collonnier et al. [13] showed that, in absence of RAD51 (*Pprad51-1-2* double mutant), CRISPR/Cas9-targeted or IR integration is increased by 30 times by an alternative end-joining reaction (alt-EJ).

*P. patens* is inherently resistant to the induction of DSB by mechanisms that either interfere with its formation or confer resistance, e.g., by rapid and robust repair [14]. The course of the DSB rejoining is best described by a two-phase exponential equation, where the course of the repair involves a fast and a slow component [15]. Genetic studies have identified as a fast component the rapid end-joining by the NHEJ pathway, while the slow component is dependent on the factors of the HR pathway [16].

For more information on HR in plant DSB repair, we investigated the involvement of two evolutionarily separated *RAD51* genes. The results show that functional *RAD51* is indispensable for the repair of DNA damage by the HR mechanism, and when absent, the entire DSB repair is carried out by a rapid NHEJ pathway. On the contrary, in the absence of *RAD51B*, HR is the preferred pathway to repair DSB. The new data also confirm our previous findings on the essential role of RAD51 in genomic stability of *P. patens* [17].

## 2. Materials and Methods

### 2.1. P. patens Materials and Cultivation

The WT *P. patens* (accession (Hedw.)) B.S.G. ‘Gransden2004’) [18], *Pprad51-1-2* double mutant [9] and the *Pprad51B* mutant (*Pprad51B#40*) were kindly provided by F. Nogué INRA, Paris, France. The *Pprad51B#40* was constructed by F. Nogué, using a CRISPR/Cas9 strategy when the nucleotides 933 to 1438 of the *RAD51B* gene have been deleted, and the line will be described in detail elsewhere. All strains were cultured either as ‘spot inocula’ on BCD agar medium supplemented with 1 mM CaCl_2_ and 5 mM ammonium tartrate (BCDAT medium) or as lawns of protonema filaments by subculturing homogenized tissue on BCDAT agar overlaid with cellophane and grown under standard conditions with 18/6 h day/night light cycle at 22/18 °C in the growth chamber [19,20]. One-day-old protonemata (1d) for repair experiments were prepared from 1-week-old tissue (7d) scraped from plate, suspended in 8 mL of BCDAT medium, sheared at 10,000 rpm for two 1-min cycles by a T25 homogenizer (IKA, Staufen, Germany), and left to recover for 24 h in a cultivation chamber, with gentle shaking at 100 rpm. This treatment yielded a suspension of 3–5 cell filaments, which readily settle for recovery. The 1d protonemata contain up to 50% of apical cells and represents dividing tissue in contrast to 7d protonemata, with only 5% of apical cells that we assumed as differentiated tissue (see [21] Appendix A).

### 2.2. Microscopic Analysis

Ten-day-old protonemata were stained with 10 µg/mL propidium iodide (PI) (Sigma-Aldrich, Schnelldorf, Germany) in liquid BCDAT medium, mounted onto a glass slide, and analyzed by a Spinning disc (SD) microscope Nikon Eclipse Ti-E, inverted (Nikon, Minato, Japan), with Yokogawa CSU-W1 SD unit (50 mm), on a Nikon Ti-E platform, Laser box MLC400 (Agilent. Santa Clara, CA, USA) Zyla cMOS camera (Andor, Belfast, Northern Ireland), Nikon Plan-Apochromat L 10 × /0.45 objective.

### 2.3. Mutagens, Treatments, and Sensitivity Assay

Sensitivity to DNA damage was measured as previously described [19] after the treatment either with freshly prepared solutions of bleomycin sulphate (bleomycin), supplied as Bleomedac inj. (Medac, Hamburg, Germany), or dilution of methyl methanesulfonate (MMS) (Sigma-Aldrich, Schnelldorf, Germany) in BCDAT medium. Protonemata were dispersed in liquid BCDAT medium containing bleomycin or MMS and treated for 1 h. After the treatment, the recovered protonemata were rinsed and inoculated as eight explants per quadrant on a Petri plate with a drug-free BCDAT agar, without the cellophane overlay, and allowed to grow under standard conditions. The treatment was assessed after 3 weeks by weighing the explants. For each experimental point, the fresh weight of the treated plants was normalized to the fresh weight of untreated plants of the same line and plotted as % of ‘Relative fresh weight’. In every experiment, the treatment was carried out in duplicate, and the experiments were repeated at least three times, so that each experimental point represents at least 48 individual plants. The results were statistically analyzed by a two-tailed Student’s *t*-test.

### 2.4. DNA Isolation and Analysis of rDNA Copy Numbers

DNA was isolated from 7-day-old protonemal tissue according to [22]. DNA quality and concentration were determined by electrophoresis in a 1% (*w*/*v*) agarose gel, using Gene Ruler 1 kb DNA (Thermo Scientific, Waltham, USA). Experimental analysis of the rDNA arrangement was performed as previously described by [17], using PCR primers Pp18S_A and Pp18S_B for *18S* rDNA and Pp_5S_F and Pp_5S_R for *5S* rDNA. The levels of rDNAs in every line were normalized to ubiquitin as a reference gene (primer combination PubqFw and PubqRev). The qPCR was performed by using qPCRBIO SyGreen Mix Lo-ROX (PCR Biosystems, London, UK) in a Stratagene RT PCR System MX3005P, La Jolla, CA, USA). For the primer sequences, see Appendix A. The data were statistically analyzed by using a two-tailed Student′s *t*-test.

### 2.5. Single-Cell Gel Electrophoresis (Comet) Assay

Repair kinetic was estimated in 1d and 7d protonemata after bleomycin or MMS treatment, as previously described [19]. Tissue was either flash-frozen in liquid N_2_ (repair t = 0) or allowed to recover in liquid BCDAT medium for the indicated repair times and then frozen. DSBs after bleomycin treatment were detected by a comet assay, using neutral N/N protocol, whereas DNA single-strand breaks (SSBs) after MMS treatment were detected with A/N protocol, which includes, after the lysis of nuclei, a treatment with alkali to reveal breaks by unwinding the DNA double helix, as described in [23,24]. Comets were stained with SYBR Gold (Molecular Probes/Invitrogen, Eugene, USA), viewed in epifluorescence with a Nikon Eclipse 800 microscope, and evaluated by the LUCIA Comet cytogenetic software (LIM Inc., Prague, Czech Republic). The fraction of DNA in comet tails (% tail-DNA) was used as a measure of DNA damage. In each experiment, the % tail-DNA was measured at seven time points, namely 0, 3, 5, 10, 20, 60, and 180 min, after the treatment and in control tissue without treatment. Measurements obtained in three independent experiments, totaling at least 300 comets analyzed per experimental point were plotted as % of remaining damage and statistically analyzed by Student’s *t*-test. Time-course repair data were analyzed for two-phase exponential decay kinetics by Prism v.5 program (GraphPad Software Inc., San Diego, CA, USA).

### 2.6. Phylogenetic Analysis

We selected representatives from paralogs of RAD51, RADA-D, DMC1, and XRCC2 and -3 [25] and aligned protein sequences by using the Clustal Omega [26] algorithm in the EMBL-EBI Web Services, with homology detection by HMM–HMM comparisons [27]. Maximum likelihood analyses of the matrices were performed in Randomized Accelerated Maximum Likelihood (RAxML 8.2.4; [28]) to examine differences in optimality between alternative topologies. One thousand replications were run for bootstrap support values. Phylogenetic trees were constructed and modified with iTOL v3.4 [29]. The sequences of RAD51 paralogs of *H. sapiens, A. thaliana, S. pombe, S. cerevisiae, P. abyssi,* and *A. fulgidus* retrieved from [7] were used for Blast search against *P. patens, Z. mays, O. sativa,* and *M. polymorpha* genome database Phytozome v13 [30]. *D. melanogaster* and *M. musculus* protein sequences were retrieved by Blast search from the National Center for Biotechnology Information https://www.ncbi.nlm.nih.gov/ (accessed on 7 November 2022 and after). An overview of the used protein sequences and their accession numbers are listed in Appendix A.

## 3. Results

### 3.1. Phylogenetic Analyses of the RecA/RAD51 Superfamily

The phylogenetic analyses (Figure 1 and extended in Appendix A) show the formation of subfamilies (a) with highly conserved functions as RAD51 and DMC1, clades B and C; (b) with relatively divergent functions, clades D, E, F, and G; and (c) clade A containing XRCC2 paralogue. Here, bacterial RecA was used as an outgroup in the plot. All main clades, A–G, are highly supported (78–100% BS).

### 3.2. Phenotypic Analysis of Pprad51B and Pprad51-1-2 Mutants

The growth of mutant lines is both reduced, as indicated by the formation of smaller colonies (Figure 2A) and their lower weight (Figure 2C), and vegetatively aberrant, as the plants develop only a few gametophores. The retardation of growth in *Pprad51B* to 80% and in *Pprad51-1-2* to 75% of WT indicates that the loss of *RAD51* affects the vegetative stage of the *P. patens* life cycle. The protonemal growth of mutant lines is also altered. While the WT protonema regularly forms lateral branches, in the mutant lines, branching is restricted, albeit in a different manner. Whereas in *Pprad51B*, side branching of filaments is already limited due to mutation, in *Pprad51-1-2*, some fibers develop side branches, but soon most of these are eliminated by cell death, and only unbranched fibers survive, as can be seen in PI-stained protonemata (Figure 2B). The lack of branching or elimination of branched filaments during the early phase of growth leads to a reduced number of leaf shoots, gametophores, at later stages of development (Figure 2A). The copy number of *18S* rDNA is slightly reduced in the *Pprad51B* and significantly reduced in the double mutant *Pprad51-1-2*, where reduction applies also to *5S* rDNA (Figure 2D).

### 3.3. Recovery from Bleomycin and MMS Treatment in Absence of RAD51^-^ and RAD51B Lines

To characterize DNA repair defects in *Pprad51B* and *Pprad51-1-2* mutants, we compared their growth responses to DSBs induced by radiomimetic drug bleomycin or to induction of small alkylation adducts added by MMS that are readily transformed into SSBs. These lesions represent blocks for DNA replication and are removed, repaired, or bypassed by various error-free (e.g., HR, base excision repair, etc.), as well as error-prone (e.g., NHEJ) pathways. The sensitivity was tested after the acute treatment for 1 h by assessing the ability of the tissue to recover from damage and grow [19]. In response to the treatment with bleomycin (Figure 3), both mutants, *Pprad51B* and *Pprad51-1-2*, manifest extreme, though almost identical, sensitivity when compared to WT (Student’s *t*-test: *p* = 0.001 and *p* = 0.001, respectively).

Contrary to bleomycin, the WT and the lines *Pprad51B* and *Pprad51-1-*2 are equally sensitive to MMS treatment (Appendix A). This observation documents that both *RAD51* genes are vitally needed for recovery from induced DSB, but evidently not for recovery from the modification of bases or induction of SSB.

### 3.4. Kinetic of DSB Repair in Pprad51B and Pprad51.1.2

DSBs were induced by 1 h treatment with 30 μg bleomycin/mL and detected by comet assay. The most visible effect of both *RAD51* genes in DSB repair is observed between dividing (1d) and differentiated (7d) tissue. In the case of 1d-regenerated lines, in which the enhanced preponderance of mitotically active apical cells, the repairing rate of DSB is the same in WT and the mutant lines *Pprad51B* and *Pprad51.1.2* (Figure 4A). Contrary to 1d, in 7d-old tissue, various roles for RAD51 and RAD51B are evident. Whereas in *Pprad51B* DSB, repair is severely inhibited, in *Pprad5-1-2*, DSBs are readily removed, even faster than in WT (Figure 4B).

The repair kinetic was evaluated by performing an analysis of the repair data with the GraphPad program Prism. Because DSB repair follows two-phase kinetics [15], a model of ‘Two-phase exponential decay’ was used to obtain values of the half-life of first (τ_fast) and second (τ_slow) phase of the repair course and the value of plateau representing the remaining unrepaired damage (Figure 5).

The half-life and plateau values in 1d tissue of WT and both mutant lines are almost identical, indicating an extremely rapid and complete repair that occurs in dividing cells, independently of RAD51. In contrast to the 1d tissue, the rate of repair in 7d tissue is much slower, revealing differences in kinetics among WT and *rad51-1-2* and *rad51B* lines. During the first phase, the DSB repair rate is two-fold faster in mutant lines than in WT, i.e., 3 and 4 vs. 8 min in *Pprad51-1-2*, *Pprad51B*, and WT, respectively. Evidently, the disruption of both *RAD51* genes accelerates the initial steps of DSB repair. Nevertheless, during the second, slow phase, while the repair of *Pprad51-1-2* versus WT is only moderately accelerated, half-life 28 vs. 32 min, the repair rate in *Pprad51B* is dramatically three-times retarded (half-life 90 min). Thus, the kinetic of the DSB repair, with both phases combined, is quickest in *Pprad51-1-2*, with the plateau of remaining damage being only 7% in contrast to 16% in WT (*p* = 0.0021) and slowest in *Pprad51B*, plateau 21% (*p* = 0.0276; see Figure 4B). MMS-induced SSBs were not repaired at all in the studied plants (Appendix A).

## 4. Discussion

*RAD51* and its paralogs, which are vital for highly evolved organisms, are not essential genes in *P. patens* [3]. Loss of *RAD51* function results in morphological abnormalities, a specific phenotype and hypersensitivity to the DSB-inducing agent bleomycin. DSB repair in *P. patens* obeys a two-phase exponential rejoining kinetic with fast and slow components [15].

The role of both studied *RAD51* genes is best seen when the repair is compared between dividing and differentiated protonemal cells. In the culture of short fragments with a majority of mitotically active apical end-cells, DSB repair is much faster than in already grown and differentiated protonemata. In WT, the repair rate of the fast phase is eight-fold quicker in dividing cells than in already differentiated cells, and similarly, four- and two-fold quicker in cells of *Pprad51B* and *Pprad51-1-2* lines, respectively. Such high repair-rates apply to cells undergoing the cell division and having active, relaxed chromatin allowing rapid NHEJ repair with no or minimal requirement for HR [16].

The DSB repair proceeds slower in differentiated cells and is sensitive, though differently, to the absence of RAD51-1 and -2 or RAD51B. DSB repair in *Pprad51-1-2* line is almost complete within 3 h, leaving only 7% damage after redirecting part of the DSBs to NHEJ or alt-EJ repair. Nevertheless, in WT and especially *Pprad51B*, the remaining damage is much higher, 16% and 21%, respectively. During the slow phase, the rate of DSB repair strongly depends on RAD51B. While the DSB half-life is similar for WT and *Pprad51-1-2* line, i.e., 32 and 28 min, respectively, for *Pprad51B*, the DSB half-life is three-fold longer, indicating a repair deficit due to the absence of RAD51B. Interestingly, in mammalian cells, elimination of ATM and BRCA2 proteins has a similar effect on the kinetic of the slow phase of DSB repair [16]. *In vitro*, the RAD51B protein and paralogs of the BCDX2 complex bind preferentially to branched DNA strands such as Holliday junctions [31]. This is the first step in a pathway from damage sensing, followed by signal transduction and HR DSB repair, in which ATM, BRCA2, and RAD51B are epistatic to each other. Our data on the kinetics of DSB repair suggest that the *P. patens* RAD51- and RAD51B- are involved in separate steps of DSB repair, with little, if any, overlap. An ancestral common feature is ssDNA recognition; however, during evolution, RAD51 and the RAD51B paralog diverged functionally, as portraited in the phylogenetic tree.

The absence of *RAD51* and *RAD51B* genes also has an impact on the copy numbers of *5S* and *18S* rDNA. While in the *Pprad51B*, the number of copies varies only slightly from WT, in the double mutant, *Pprad51-1-2* is significantly reduced to 50% of the WT. This suggests that when RAD51 is missing, deletion-prone *RAD51*-independent repair pathways prevail as, for example, single-stranded annealing or NHEJ that are both assumed to be active during early stages of DSB repair. If the single-strand annealing or alt-EJ becomes the primary pathway to suppress DSBs in *Pprad51-1-2*, this may result in a higher frequency of rDNA unit deletions, unlike *Pprad51B*, in which outcome of DSB repair is close to WT. This possibility is supported by a preference for rapid DSB repair in *Pprad51-1-2*, but not in WT and *Pprad51B*.

The genomic instability of the *RAD51* double mutant is manifested in the developmental phenotype by increased induction of cell death in growing filaments, preferably those that branch, as seen in *Pprad51-1-2*. The end effect is growth retardation and the formation of fewer gametophores on explant colonies; however, a similar end effect was also observed in *Pprad51B*, with reduced branching of filaments, and this also leads to the formation of fewer gametophores as a consequence of missing RAD51B.

The primary function of the RAD51 is to promote and execute HR, an important activity in the life of all eukaryotes, from yeast to man. During evolution, the ancestral protein segregated into a large family of proteins that mostly share a common structural framework but diverge substantially in their involvement in HR. RAD51 and DMC1 are close recombinases, which, during evolution, separated from other ancestorial RAD51s. RAD51 paralogs A, B, C, and D; XRCC2; and XRCC3 evolved into separate group, and RAD51B, RAD51C, RAD51D, XRCC2 were identified in *P. patens*. While RAD51 and DMC1 are core nucleoprotein filament formation proteins for strand invasion, the function of the paralogs in HR is less understood than, for instance, RAD51B of *P. patens*, which differs from RAD51 in many ways [12].

The two copies and intron-less structure of *RAD51* in *P. patens* suggest its unique position in this organism, probably to backup and secure the presence of its function. One can speculate about its roles, for instance, in accelerating evolutionary diversity by promoting processes associated with HR, such as gene duplication or reconstruction to adapt the organism to new challenges. In the case of *P. patens*, it could be the transition of moss from water to soil, and this might be the main gain from evolving such an intricate structure of *RAD51*, which is a non-essential gene of *P. patens*.

## Figures and Tables

**Figure 1 genes-14-00305-f001:**
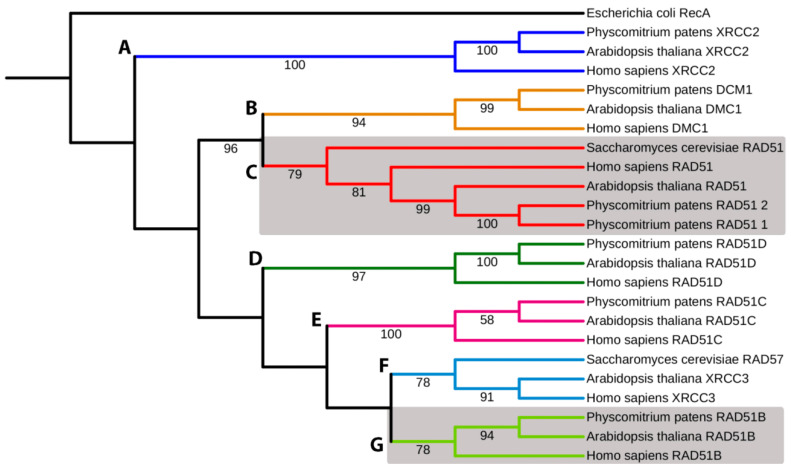
Simplified phylogenetic tree of RAD51 superfamily. The evolutionary history was inferred by using maximum likelihood analysis. RecA *Escherichia coli* was used as the outgroup. Bootstrap support values > 50% are shown on the branch above. The ML log likelihood is −16,689.642127. The analysis included 24 amino acid sequences and 673 positions in the final data set. (A) XRCC2, (B) DCM1, (C) RAD51-1 and -2, (D) RAD51D, (E) RAD51C, (F) XRCC3, and (G) RAD51B. Gray boxes depict RAD51-1 and -2 and RAD51B clads. Accession numbers are listed in Appendix A.

**Figure 2 genes-14-00305-f002:**
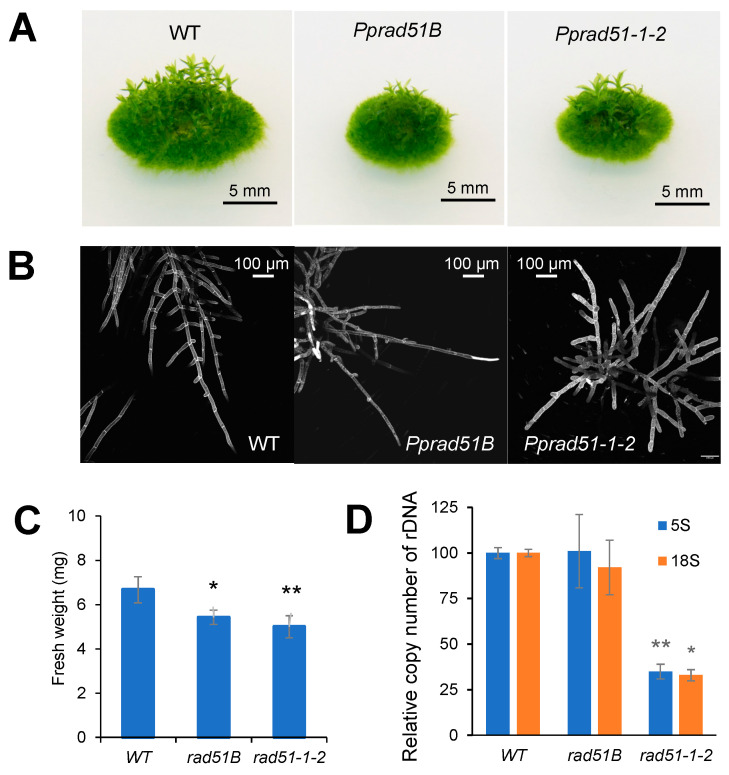
Characterization of moss *Pprad51s* mutant lines. (**A**) Morphology of 1-month-old colonies of WT and *Pprad51B* and *Pprad51-1-2* mutant lines grown on BCDAT medium without bleomycin treatment. (**B**) Seven-day-old protonemata stained with propidium iodide. (**C**) The growth rates of WT and mutant lines measured as fresh weight of 3-week-old untreated plants. (**D**) Loss of *5S* and *18S* rDNA genomic copies in WT and mutant lines. Relative rDNA copy numbers were measured by qPCR and normalized to values obtained in the WT. Student’s *t*-test: * *p* < 0.05; ** *p* < 0.01. Error bars represent SD.

**Figure 3 genes-14-00305-f003:**
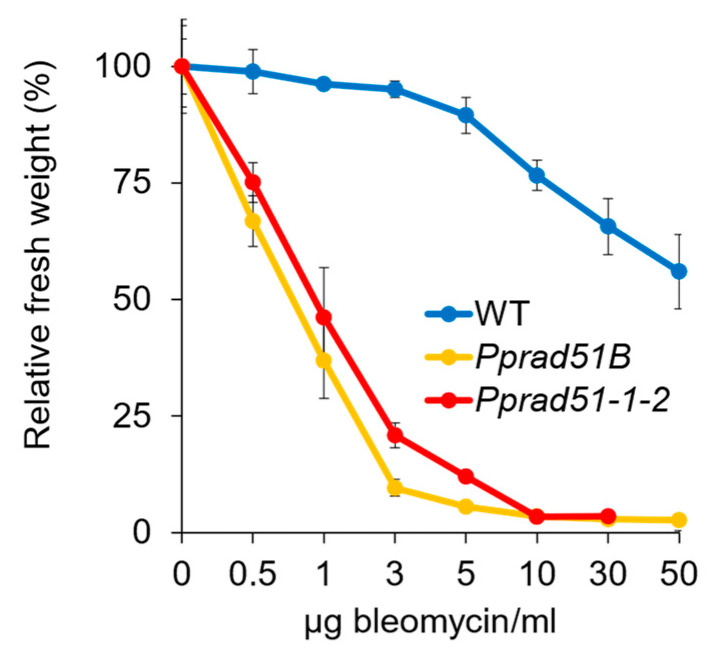
Growth response of WT, *Pprad51B*, and *Pprad51-1-2* plants to 1 h treatment with 0.5, 1, 3, 5, 10, 30, and 50 μg bleomycin/mL. After the treatment, explants were inoculated on Petri plates with drug-free BCDAT medium and grown under standard conditions for 3 weeks. For each experimental point, the weight of treated plants collected from two replicas was normalized to the weight of untreated plants and plotted as relative fresh weight, which was set as a default to 100 for untreated plants. Error bars represent SD.

**Figure 4 genes-14-00305-f004:**
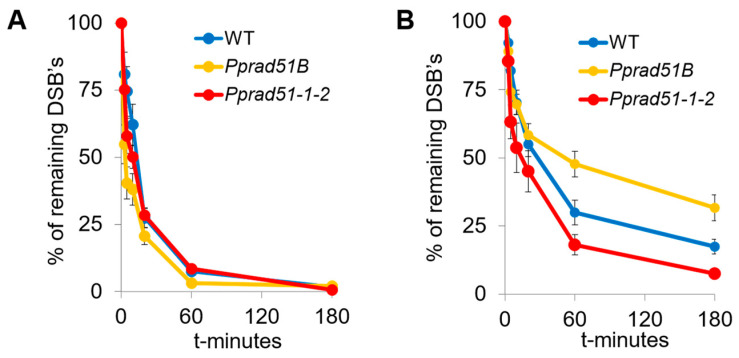
DSB repair kinetics determined by comet assay in dividing (1d) or differentiated (7d) tissue of WT (blue), *Pprad51B* (orange), and *Pprad51-1-2* (red) mutant lines. Protonemata, which regenerated for 1 (**A**) or 7 (**B**) days after subculture were treated with 30 μg bleomycin/mL for 1 h. Repair kinetics was measured as % of DSBs remaining after the 0, 3, 5, 10, 20, 60, and 180 min of repair recovery. Maximum damage is normalized as 100% at *t* = 0 for all lines. Error bars indicate SD.

**Figure 5 genes-14-00305-f005:**
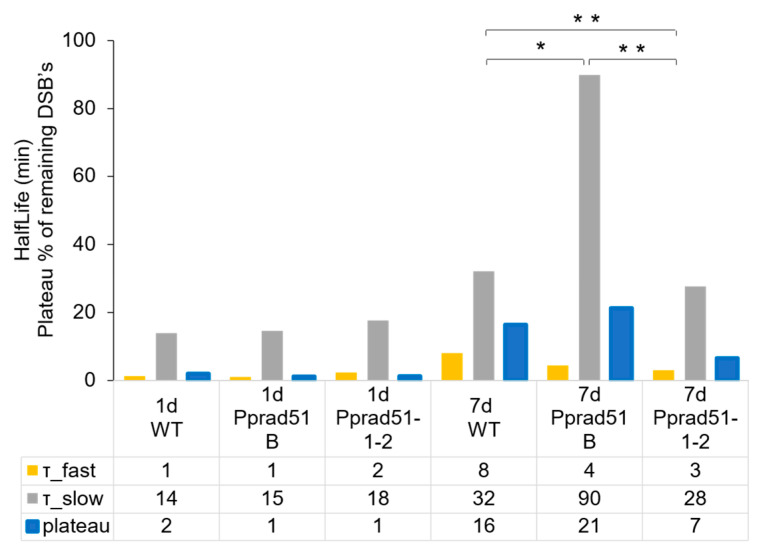
Analysis of DSB repair data. Plotted are the parameters of the ‘Two phase decay’ kinetics model that were determined by GraphPad program Prism v.5. The half-life of first (τ_fast) and second (τ_slow) phase of the repair curve and the plateau of remaining damage are plotted in orange, gray, and blue, respectively. The numeric values are provided in the table below the plot. Student’s *t*-test: * *p* < 0.05; ** *p* < 0.01.

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
