# Peer review of "RAD51 and RAD51B Play Diverse Roles in the Repair of DNA Double Strand Breaks in Physcomitrium patens"

_genes, 2023, doi:10.3390/genes14020305_

Round 1

Reviewer 1 Report

This work, to understand the roles of RAD51 and its paralogs in P. patens is of high significance but the experiments involved didn’t elucidate the functional importance of these proteins. Though the data show that these proteins are involved in the DSB repair pathway, their significance is not explained in this study.

There are a lot of typographical errors in the manuscript. They should be addressed properly.

Line 210 and Fig S2: The MMS concentrations used seem too high to distinguish effects of mutants and WT cell lines. An experiment with lower MMS concentrations should be done as well.

The kinetics of DSB repair data analysis: This part is confusing. The assumption here is that the fast phase of the DSB repair kinetics is via NHEJ pathway and the slow phase is via HR pathway. The results for the 7d cells indicated the fast kinetics changed between WT and RAD51-1-2 mutant. But the slow kinetics are identical. The conclusion stated was that there is a shift from HR to NHEJ pathway in this mutant. But if this is the case then why are the slow kinetics identical? If the mutant is HR-defective, then this rate will be different. I believe the data does show some alternate pathway might be involved to bypass HR in this mutant, but the 2-state analysis doesn’t back up the claim. There must be some other control experiments done to show NHEJ is more active in this mutant. Currently, it is not clear.

Author Response

Reviewer 1

This work, to understand the roles of RAD51 and its paralogs in P. patens is of high significance but the experiments involved didn’t elucidate the functional importance of these proteins. Though the data show that these proteins are involved in the DSB repair pathway, their significance is not explained in this study.

A: Newly addressed in Discussion-

There are a lot of typographical errors in the manuscript. They should be addressed properly.

A: All identified and by Reviewer 2 indicated typographic errors were corrected in manuscript.

Line 210 and Fig S2: The MMS concentrations used seem too high to distinguish effects of mutants and WT cell lines. An experiment with lower MMS concentrations should be done as well.

A: Statistic evaluation for both bleomycin and MMS treatments were added to support statements in the ms.

The kinetics of DSB repair data analysis: This part is confusing. The assumption here is that the fast phase of the DSB repair kinetics is via NHEJ pathway and the slow phase is via HR pathway. The results for the 7d cells indicated the fast kinetics changed between WT and RAD51-1-2 mutant. But the slow kinetics are identical. The conclusion stated was that there is a shift from HR to NHEJ pathway in this mutant. But if this is the case then why are the slow kinetics identical? If the mutant is HR-defective, then this rate will be different. I believe the data does show some alternate pathway might be involved to bypass HR in this mutant, but the 2-state analysis doesn’t back up the claim. There must be some other control experiments done to show NHEJ is more active in this mutant. Currently, it is not clear.

A: I agree the text was confusing. The whole paragraph is reformulated. The shift from HR to NHEJ or alt-EJ in Pprad51-1-2 line was already proved earlier by sequence analysis of IR or CRISP/Cas9 induced integration sites by Collonnier et al. Reference is added to Introduction.

Reviewer 2 Report

Manuscript entitled ”RAD51 and RAD51B play diverse roles in the repair of DNA double strand breaks in Physcomitrium patens” by Angelis et al describes a comparative study of two knock-out lines of P. patens in RAD51 and RAD51B genes, that code for proteins involved in DNA repair mechanisms belonging to the  RecA/RAD51 superfamily. The authors perform a Phenotypic analysis of WT, single mutant Pprad51B and double mutant Pprad51-1-2 lines of P. patens in and compare their susceptibility to double strand break mutagen agents (bleomycin and MMS) as well as their DNA repair kinetics.  The experimental design and the proposed approaches used in this work seem both correct for the most part. Also, the results obtained are interesting, However, in my opinion, some aspects of the manuscript need to be revised, particularly the discussion. Some of them are enumerated in the additional comments below:

As a general comment, the discussion of work in this version of the manuscript is often rather superficial and should be improved. Although the authors have performed a good interpretation of the results obtained, the discussion lacks an integration into previous results from the literature that could help fit their observations in a more robust framework.

Additional comments:

a) Regarding Introduction:

Line 28: RAD51 should appear without italics (the sentence refers to the protein)

Line 58: NHEJ abbreviature should be described here

Regarding Materials and Methods:

Line 74:  Rensing et al., 2020 should be referenced appropriately

Line 75:  Schaefer et al. 2010 should be referenced appropriately

Line 80: please change “1mM” and “5mM” with “1 mM” and “5 mM”

Line 92: please change “10μg/ml” with “10 μg/ml”

Line 95: please change “50mm” with “50 mm”

Line 113: please change “1%(w/v)” with “1 % (w/v)”

Line 114: please change “discribed” with “described”

Line 119: please change “Frimer sequences” with “Primer sequences:”

Line 120: please change “The data were” with “The data was”

Line 126: SSB abbreviature should be defined

Lines 136-137: please change “Time-course repair data were analyzed” with “Time-course repair data was analyzed”

b) Regarding Results:

Lines 180-182 and Figure 4D: Although the result is very clear, Figure 4D data needs statistical analysis to confirm a significant reduction in copy number of 18S rRNA in the double mutant

Lines 218-219: please consider changing “the repair of DSBs is identically rspid in WT as well as both mutant lines Pprad51B and Pprad51.1.2” with “the repairing rate of DSBs is the same in WT and both mutant lines Pprad51B and Pprad51.1.2”

c) Regarding supplemental data:

Figure S2 legend: “(A) Growth response to MMS. Explants of 7d protonemata were treated for 1 hr with 10, 30 and 50 mM MMS. After treatment, explants were inoculated on Petri plates with drug-free BCDAT medium and weighted after 3 weeks of growth (see legend for Figure 3 in main text)”

Author Response

Reviewer 2

Manuscript entitled ”RAD51 and RAD51B play diverse roles in the repair of DNA double strand breaks in Physcomitrium patens” by Angelis et al describes a comparative study of two knock-out lines of P. patens in RAD51 and RAD51B genes, that code for proteins involved in DNA repair mechanisms belonging to the  RecA/RAD51 superfamily. The authors perform a Phenotypic analysis of WT, single mutant Pprad51B and double mutant Pprad51-1-2 lines of P. patens in and compare their susceptibility to double strand break mutagen agents (bleomycin and MMS) as well as their DNA repair kinetics.  The experimental design and the proposed approaches used in this work seem both correct for the most part. Also, the results obtained are interesting, However, in my opinion, some aspects of the manuscript need to be revised, particularly the discussion. Some of them are enumerated in the additional comments below:

As a general comment, the discussion of work in this version of the manuscript is often rather superficial and should be improved. Although the authors have performed a good interpretation of the results obtained, the discussion lacks an integration into previous results from the literature that could help fit their observations in a more robust framework.

A: Part of Discussion related to DSB repair was rewritten. 

Additional comments:

A: The authors thank the reviewer for the detailed marking of typographical errors and correction of clumsy wording.

  1. a) Regarding Introduction:

Line 28: RAD51 should appear without italics (the sentence refers to the protein)

A: Corrected

Line 58: NHEJ abbreviature should be described here

A: Added

Regarding Materials and Methods:

Line 74:  Rensing et al., 2020 should be referenced appropriately 

Line 75:  Schaefer et al. 2010 should be referenced appropriately

A: Both references corrected

Line 80: please change “1mM” and “5mM” with “1 mM” and “5 mM”

A: Corrected

Line 92: please change “10μg/ml” with “10 μg/ml”

A: Corrected

Line 95: please change “50mm” with “50 mm”

A: Corrected

Line 113: please change “1%(w/v)” with “1 % (w/v)”

A: Corrected

Line 114: please change “discribed” with “described”

A: Corrected

Line 119: please change “Frimer sequences” with “Primer sequences:”

A: Corrected

Line 120: please change “The data were” with “The data was”

A: Corrected

Line 126: SSB abbreviature should be defined

A: Added

Lines 136-137: please change “Time-course repair data were analyzed” with “Time-course repair data was analyzed”

A: Changed

  1. b) Regarding Results:

Lines 180-182 and Figure 4 2D: Although the result is very clear, Figure 4 2D data needs statistical analysis to confirm a significant reduction in copy number of 18S rRNA in the double mutant

A: Statistical evaluation added in Figure 2D

Lines 218-219: please consider changing “the repair of DSBs is identically rspid in WT as well as both mutant lines Pprad51B and Pprad51.1.2” with “the repairing rate of DSBs is the same in WT and both mutant lines Pprad51B and Pprad51.1.2”

A: Changed

  1. c) Regarding supplemental data:

Figure S2 legend: “(A) Growth response to MMS. Explants of 7d protonemata were treated for 1 hr with 10, 30 and 50 mM MMS. After treatment, explants were inoculated on Petri plates with drug-free BCDAT medium and weighted after 3 weeks of growth (see legend for Figure 3 in main text)”

A: Incorporated

Round 2

Reviewer 1 Report

The edits have greatly improved the manuscript. Most of the issues have been addressed.

I still would like a comment about why for the RAD51-1-2 mutant, the HR slow kinetics rate is similar to WT. For RAD51 deletion, HR should be defective and the rate should change.

For the RAD51B mutant discussion, it is stated that the study reveals RAD51 and RAD51B are involved in different pathways. I agree that the data shows the impact of RAD51B on HR kinetics but it doesn't show it is involved in a different pathway. I think little rephrasing is required.

Also, some places have 'RAD51' and some have 'Rad51'. Please try to be consistent.

Author Response

The edits have greatly improved the manuscript. Most of the issues have been addressed.

I still would like a comment about why for the RAD51-1-2 mutant, the HR slow kinetics rate is similar to WT. For RAD51 deletion, HR should be defective and the rate should change.

A: I agree that RAD51-dependent HR should be defective in the absence of RAD51, in P. patens of both RAD51-1 and 2. However, Figures 4 and 5 show the actual removal of DSBs detected directly by the comet assay, not the mechanism by which this was achieved. The involvement of NHEJ or alt-EJ in Pprad51-1-2 was only proven by subsequent analysis of the insertion sites.

For the RAD51B mutant discussion, it is stated that the study reveals RAD51 and RAD51B are involved in different pathways. I agree that the data shows the impact of RAD51B on HR kinetics but it doesn't show it is involved in a different pathway. I think little rephrasing is required.

A: I agree, it is overstated. Pathways replaced by rather neutral “steps of DSB repair”

Also, some places have 'RAD51' and some have 'Rad51'. Please try to be consistent.

A: Corrected

Reviewer 2 Report

I am happy to see that the authors have addressed most of the issues pointed out before, and that the revised version of the manuscript has improved with the changes made, particularly the discussion. Still, I feel that several minor issues should be addressed:

Line 92: please change “[19] [20]” with “[19,20]”

Line 97: please change “50%” with “50 %”

Line 98: please change “5%” with “5 %”

Line 170: please change “(78-100% BS)” with “(78-100 % BS)”

Line 174: please change “>50%” with “> 50 %”

Lines 182-183: please change “to 80% and in Pprad51-1-2 to 75% of WT” with “to 80 % and in Pprad51-1-2 to 75 % of WT” “

Line 204: I am not sure about the use of the terms “RAD51- and RAD51B-“ referring about the mutant lines. Perhaps is more clear  Pprad51B and Pprad51.1.2 mutant lines” or simply “in absence of  RAD51 and RAD51B”

Line 243: please change “100%” with “100 %”

Line 292: Line 98: please change “7%” with “7 %”

Line 304: I think that the expression “that the RAD51- and RAD51B-“is not correct, because you are talking about the proteins (and  not the deletion mutants, that as a mentioned before, I do not think that this should be the correct way of naming them). Perhaps is better “that P. patens  RAD51- and RAD51B “ (your work only has showed data about these proteins in P. patens)

Line 319: RAD51 should appear in italics (the sentence refers to the gene)

Author Response

I am happy to see that the authors have addressed most of the issues pointed out before, and that the revised version of the manuscript has improved with the changes made, particularly the discussion. Still, I feel that several minor issues should be addressed:

Line 92: please change “[19] [20]” with “[19,20]”

A: Changed

Line 97: please change “50%” with “50 %”

A: Changed

Line 98: please change “5%” with “5 %”

A: Changed

Line 170: please change “(78-100% BS)” with “(78-100 % BS)”

A: Changed

Line 174: please change “>50%” with “> 50 %”

A: Changed

Lines 182-183: please change “to 80% and in Pprad51-1-2 to 75% of WT” with “to 80 % and in Pprad51-1-2 to 75 % of WT” “

A: Changed

Line 204: I am not sure about the use of the terms “RAD51- and RAD51B-“ referring about the mutant lines. Perhaps is more clear  “Pprad51B and Pprad51.1.2 mutant lines” or simply “in absence of  RAD51 and RAD51B”

A: Used “simply”, it was what I meant ?

Line 243: please change “100%” with “100 %”

A: Changed

Line 292: Line 98: please change “7%” with “7 %”

A: Changed

Line 304: I think that the expression “that the RAD51- and RAD51B-“is not correct, because you are talking about the proteins (and  not the deletion mutants, that as a mentioned before, I do not think that this should be the correct way of naming them). Perhaps is better “that P. patens  RAD51- and RAD51B “ (your work only has showed data about these proteins in P. patens)

A: Correct, added P. patens

Line 319: RAD51 should appear in italics (the sentence refers to the gene)

A: Reformatted
